# Quantitative Detection of Pipeline Cracks Based on Ultrasonic Guided Waves and Convolutional Neural Network

**DOI:** 10.3390/s24041204

**Published:** 2024-02-13

**Authors:** Yuchi Shen, Jing Wu, Junfeng Chen, Weiwei Zhang, Xiaolin Yang, Hongwei Ma

**Affiliations:** 1Department of Civil Engineering, Qinghai University, Xining 810016, China; sycamore007s@outlook.com (Y.S.);; 2Department of Mechanical Engineering, Dongguan University of Technology, Dongguan 523808, China; 3School of Mechanics and Construction Engineering, Jinan University, Guangzhou 510632, China; 4Guangdong Provincial Key Laboratory of Intelligent Disaster Prevention and Emergency Technologies for Urban Lifeline Engineering, Dongguan 523808, China

**Keywords:** convolutional neural network, ultrasonic guided wave, pipeline, crack defects

## Abstract

In this study, a quantitative detection method of pipeline cracks based on a one-dimensional convolutional neural network (1D-CNN) was developed using the time-domain signal of ultrasonic guided waves and the crack size of the pipeline as the input and output, respectively. Pipeline ultrasonic guided wave detection signals under different crack defect conditions were obtained via numerical simulations and experiments, and these signals were input as features into a multi-layer perceptron and one-dimensional convolutional neural network (1D-CNN) for training. The results revealed that the 1D-CNN performed better in the quantitative analysis of pipeline crack defects, with an error of less than 2% in the simulated and experimental data, and it could effectively evaluate the size of crack defects from the echo signals under different frequency excitations. Thus, by combining the ultrasonic guided wave detection technology and CNN, a quantitative analysis of pipeline crack defects can be effectively realized.

## 1. Introduction

Owing to its evident advantages, including minimal environmental impacts, large carrying capacities, high efficiencies, and low costs, pipeline transportation is expected to play an increasingly important role in future transportation engineering [1]. However, in pipeline engineering, the integrity and safety of pipeline structures form the premise of smooth transportation. In their absence, a loss of transported materials caused by damaged pipeline structures is expected to not only affect production and daily life, causing huge economic losses, but also exert a huge impact on the environment, even leading to major safety risks such as collapse and explosion [2,3]. Therefore, the nondestructive testing of pipeline structures is of considerable significance. In this regard, ultrasonic guided wave technology is widely used in long-distance pipeline inspection owing to its advantages, such as easy excitation, a long propagation distance, wide coverage, high detection accuracy, and low detection cost [4]. However, owing to the complex characteristics of ultrasonic guided waves, such as multimode and dispersion, and the influence of noise, quantitative evaluations of defect echoes in guided wave signals represent a significant challenge [5,6].

Interestingly, the majority of previous research on ultrasonic guided waves focuses on the localization and qualitative analysis of defects, relying on the extraction of a series of characteristic parameters from the original signals [7,8]. By contrast, quantitative research on defects directly based on the original guided waves is rather limited. In particular, only a few quantitative studies have been conducted on defects by directly using the original guided wave signals. Zhan et al. [7] used a deep learning approach to classify and detect pipe welds in noisy environments. Li et al. [8] used the CNN-LSTM hybrid model to classify pipeline defects. This research has proposed a solution for classifying pipeline defects, but it has not thoroughly explored the quantification of pipeline damage. Quantifying pipeline damage often requires a large amount of data to perform a regression analysis across the full range of damage. Determining its impact on simulation and experimental data regarding working condition diversity is a considerable challenge. Davies et al. [9] analyzed the relationship between defect sizes of cracks and small apertures and the defect echo amplitude when evaluating synthetic path image focusing and source imaging methods. Zheng [10] used a matching pursuit algorithm to quantitatively analyze the axial defect size of a pipeline. Both studies employed quantitative methods based on theoretical calculations, which may be subject to errors after environmental changes. Neural networks trained on real-time data can learn to adapt to specific environmental situations with greater compatibility. Li [11] utilized a 2D blind convolution method to estimate the dimensions of axial defects using multiple sets of data obtained from an axial sensor array. Li [12] proposed a quantitative reconstruction method for ultrasonic guided wave defects based on deep learning. This was achieved by combining the theoretical method of shear wave quantitative reconstruction of plate thinning defects, wave number space domain transform, and a convolutional neural network (CNN) using local fusion. Acciani et al. [13] used wavelet transforms and neural networks to quantitatively evaluate pipe surface damage. Preprocessing ultrasonic guided wave signals is necessary for these methods, which may exclude significant information from the original signals. A CNN network that extracts information directly from the ultrasound-guided wave response signal is highly effective in avoiding this issue. Huang [14] proposed a damage detection method based on a CNN-LSTM network for laser ultrasonic guided wave scanning detection. Miorelli et al. [15] proposed an automatic method for localizing and quantifying structural health monitoring defects based on guided wave imaging by combining convolutional neural networks. Yin et al. [16] automated the detection of pipeline defects using closed-circuit television (CCTV) and deep learning. Both studies relied on structural damage imaging and used training samples directly derived from 2D images. This method is computationally demanding and relies on image-based recognition, which can make predicting results challenging due to the effects of image imaging. The use of CNNs for quantitative defect identification can effectively prevent the masking of critical information and reduce the technical difficulty of engineering applications.

Neural network-based algorithms have made it more convenient, efficient, and accurate to quantitatively identify structural damage using ultrasonic guided wave technology. With this background, this paper proposes a method for the end-to-end quantitative characterization of pipeline defects by directly inputting the original ultrasonic guided waves signal into a network model without any preprocessing. The goal of the method is to predict the angles of radial defects on the pipe by regressing the angles of the defects directly to the prediction. To achieve this task, ceramic piezoelectric sheets are used to line a section of the pipe symmetrically at equal intervals for the excitation and reception of ultrasonic guided waves. The received signals are then used as inputs for a neural network for feature learning. The size of the penetrating crack in the pipeline is quantified using the defect damage angle, and the amount of crack damage in the ultrasonic guided wave signal is determined using a neural network method. By arranging the ceramic piezoelectric sheet symmetrically and in the same direction on the pipe, it is possible to excite ultrasonic guided waves of L mode. This mode is more sensitive to circumferential cracks in the pipe, making it better suited for the quantitative characterization of such cracks.

## 2. Defect Quantification Method Based on Ultrasonic Guided Waves and CNN

An Artificial Neural Network (ANN) represents a type of simulation and approximation of a biological neural network. It is an adaptive nonlinear dynamic network system composed of a large number of neurons connected through mutual connections, and it is primarily composed of input, hidden, and output layers [17]. The most basic unit of an ANN is a neuron, which can be expressed as Equation (1).
(1)$$y=f(\omega_{1}x_{1}+\omega_{2}x_{2}+\omega_{3}x_{3}\ldots \ldots \omega_{n}x_{n}+b)$$
where $x_{i}$ denotes the input n features in a neuron, $\omega_{i}$ denotes the weight value of the input feature $x_{i}$ connected to the neuron, b denotes the internal bias of the neuron, y denotes the output value of the neuron, and f(……) denotes the activation function. The more common activation functions include the rectified linear unit (ReLU) [18], Sigmoid, Tanh(x), and radial basis functions [19]. According to the findings of Acciani [11], the ReLU function is directly selected as the activation function for this study.

A CNN is a typical deep learning method developed in recent years, with wide applications in fields such as pattern recognition and medical engineering [20]. The basic structure of a CNN consists of input, convolutional, pooling, fully connected, and output layers [21]. Because each neuron of the output feature surface in the convolutional layer is locally connected to its input, and the input value of the neuron is obtained based on the weighted sum of the corresponding connection weight and the local input plus the bias value, this process is equivalent to the convolution process, from which its name is derived [22]. CNNs are primarily used for the feature recognition of two-dimensional images, whereas 1D-CNNs have only one dimension; therefore, they are widely used for feature recognition and time-series extraction. Although a 1D-CNN only has a single dimension, it demonstrates the same advantages as a CNN [14]. Specifically, CNNs not only present the advantages of traditional neural networks, such as a strong self-learning ability and good adaptability, but also the advantages of weight sharing and easy model training [23]. This study employs a CNN model, illustrated in Figure 1, which comprises an input layer, two convolutional and pooling layers, and two fully connected and output layers.

In order to compare the prediction performance of different types of neural networks, this study architects a multilayer perceptron (MLP) neural network model, as shown in Figure 2. It consists of an input layer of 1000 sets of guided wave signals, a first hidden layer of 128 neurons, a second hidden layer of 64 neurons, and an output layer.

The neural networks were trained using computers equipped with R9-5900HS processors and RTX-3050ti graphics cards to prevent external factors from affecting the detection results. The neural networks were trained using several program libraries, including Pytorch, NumPy, Pandas, and Scikit-learn. Additionally, the Adaptive Moment Estimation optimization algorithm was utilized.

The convolution layer performs a valid convolution operation using a 3 × 1 kernel, and the convolution operation is like a mathematical operation on two functions, which produces the mapping relationship of the third function, and its mathematical expression is Equation (2). f and g represent two different mapping functions. The data pass through two convolutional layers and two fully connected layers before outputting the final prediction.
(2)$$\left(f\cdot g\right)\left[n\right]=\sum_{m=-\infty }^{\infty }f\left[n-m\right]g[m]$$

The time-domain signals of ultrasonic guided waves were used as direct inputs in our quantitative characterizations of pipeline crack defects. The input signal range included the excitation wave, defect echo, and first-end face echo. In the numerical simulation stage, the displacement signal within this period was directly specified in the analysis step, and 1000 groups of amplitude signals were collected as data. During the acquisition of experimental data, the oscilloscope was set to a sampling rate of 5 M/S, resulting in a total of 10 K sampled points. To decrease the calculation time of the neural network model, each experimental signal was divided into seven data groups, each with a length of 1000. The division started from the 800th point, which was where the excitation of ultrasonic guided wave signals appeared, and ended at the 8000th point, which was the end of the first end-face echo, as shown in Figure 3. This method can increase the diversity of the data as much as possible on the basis of extracting real guided wave information.

The construction of the 1D-CNN in this paper required the following hyperparameters to be determined: the number of nodes in the hidden layer, the batch size for testing, the random seeds, the learning rate, and the number of training generations. For the selection of the number of hidden nodes, empirical Formulas (3)–(5) were used to calculate and set the number of hidden nodes with the best results. n is the number of nodes in the input layer; l is the number of nodes in the output layer; *m* is the number of hidden nodes; and *a* is a constant between 1 and 10. The remaining hyperparameters were determined through empirical formulas, which was carried out near the empirical values, and the optimal training results were selected as the parameters.
(3)$$m=\sqrt{n+l}+a$$
(4)$$m={log}_{2}n$$
(5)$$m=\sqrt{nl}$$

Determining the hyperparameters is crucial for evaluating network performance. Excellent hyperparameters correspond to excellent performance indicators and determine the success of the network model.

To set the hyperparameters of the neural network model, certain tricks for modulating these hyperparameters were obtained from the literature [24]. The neural network model architecture adopted a structure of 2–3 hidden layers, and ReLU was used as the activation function. For the simulation and experimental data, a multi-layer perceptron (MLP) and CNN were used for training and comparison, respectively. The root mean square error (RMSE), mean absolute percentage error (MAPE), and coefficient of determination (R-square) were used to comprehensively evaluate the regression performance of the neural networks. Note that the RMSE and MAPE reflect the error between the predicted and actual values of the network. The RMSE increases with larger errors, while the MAPE decreases. The R-square value indicates the quality of the model fit; the closer it is to one, the better the fit. Based on these parameters, the advantages and disadvantages of the two different neural networks in the quantitative prediction of pipeline defects were comprehensively compared. The expressions for the above parameters are shown in (6), (7), and (8). Here, $\hat{y}$ represents the true value, y represents the predicted value, and *n* represents the number of test samples.
(6)$$RMSE=\sqrt{\frac{1}{n}\sum_{i=1}^{n}{\left(y_{i}-{\hat{y}}_{i}\right)}^{2}}$$
(7)$$MAPE=\frac{100\%}{n}\sum_{i=1}^{n}\left|\frac{{\hat{y}}_{i}-y_{i}}{y_{i}}\right|$$
(8)$$R^{2}=1-\frac{\sum_{i}{\left({\hat{y}}_{i}-y_{i}\right)}^{2}}{\sum_{i}{\left({\bar{y}}_{i}-y_{i}\right)}^{2}}$$

## 3. Numerical Simulation

### 3.1. Pipeline Modeling

The JMatPro (Version 7.0) and Abaqus (Version 6.16-1) software were used for the numerical simulations. The pipe used in the experiment was a 304 seamless steel pipe with a length of l=3 m, an outer diameter of d=60 mm, and a pipe wall thickness of t=2 mm. To ensure consistency between the simulation and experiment, the JMatPro software was used to predict the performance parameters of the pipeline material to be selected. The density of the material used in the experiment was obtained as $\rho =7.855\;{g/cm}^{3}$, Young’s modulus was obtained as E=211.3 GPa, Poisson’s ratio was obtained as ν=0.2969, and the data corresponding to the material performance parameters were input into the Abaqus software. Simultaneously, a pipe model with a length of 3 m, an outer diameter of 60 mm, and a wall thickness of 2 mm was established using the Abaqus software, and the dynamic display analysis method was used to perform the calculation.

To conveniently select the central frequency of the excitation signal, the dispersion curve of the simulated pipeline was drawn using the Disperse (Version 2.0.16c) software, as depicted in Figure 4. According to the dispersion curve, the group velocity and phase velocity of the guided wave signal tended to be stable in the L (0, 2) mode near the central frequency of 70 kHz, and the wave velocity was approximately 5.5 km/s; hence, an excitation signal with a central frequency of 70 kHz was used to generate a single-mode guided wave.

In this study, a hexahedral structured mesh was used for the non-defective part of the pipeline, and a hexahedral swept mesh was used for the defective part. Generally, to control the propagation error of a waveform, the presence of multiple elements in one wavelength is essential, and the grid size of the axial elements should satisfy the conditions in Equation (9).
(9)$$l_{e}<\frac{min\left(C_{p}\right)}{10\times f}$$
where:

$l_{e}$: the axial grid cell length;

$C_{p}$: the phase velocity of the guided wave;

f: the frequency of the guided wave.

Based on the calculations, the size of the largest grid axial element was found to be approximately 7 mm. To ensure that the number of grids in the global mesh generation was an integer, the global mesh size was finally determined to be 5 mm. Considering that the gradient of the lateral section damage angle growth in the numerical simulation was one, and the increment of its reaction in the mesh size was approximately 0.5 mm, the mesh size was locally encrypted to 0.5 mm in the defective part (Figure 5).

To facilitate grid division when setting defects, the lateral section damage angle was adopted as the defect measurement index (Figure 6). For this, a pipeline model with a lateral section damage angle of 0–180° was established with a gradient of one. In total, we had 181 models. Notably, this type of model only considers penetrating cracks in the pipeline. Considering the pipeline loss along the crack-depth direction, the number of models can be increased in multiple ways. Meanwhile, considering the fact that the crack depth of non-penetrating cracks cannot be accurately measured and controlled in the experimental stage, the quantification of non-penetrating cracks was not considered in this study.

To ensure that the complete waveform image, including the excitation wave, defect echo, and first-end face echo, could be observed in the analysis time, the analysis step time was required to meet the conditions in Equation (10).
(10)$$T_{s}>\frac{2L}{min\left(C_{g}\right)}$$
where:

$T_{s}$—the analysis step time;

L—the length of the pipe;

$C_{g}$—the group velocity of the guided wave.

### 3.2. Excitation Signal

The excitation signal was a 10-cycle sinusoidal signal modulated by a cosine function with a central frequency of 70 kHz, as depicted in Figure 7. This signal energy appeared to be more concentrated near the central frequency, which is conducive to signal identification. The signal function expression is given in Equation (11) [25].
(11)$$f\left(t\right)=0.5\left[1-cos\left(\frac{2\pi f_{c}t}{n}\right)\right]sin(2\pi f_{c}t)$$
where:

$f_{c}$—the center frequency of the excitation signal;

n—the number of cycles.

### 3.3. Defect Quantification with ANN

The numerical simulation results indicated the presence of 181 groups of guided wave signals. The data were divided into training, testing, and validation sets in a ratio of 6:2:2. As stated, a 1D-CNN was used to learn these data and compare the performance with the traditional MLP network. To reduce the training time of the network, the entire guided wave signal was arranged in 1000 groups of characteristic values to be input into the network according to the time sequence, and the damage angle of the crack section corresponding to the guided wave signal was considered as an output value. The RMSE, MAPE, and R-square values were used to evaluate the regression performance of the neural networks.

As stated, a 1D-CNN was used to learn the simulation data, and the ReLU function was used as the activation function. The loss curves and training results are indicated in Figure 8. The results for the regression performance indicators of the model are as follows: RMSE = 0.948, MAPE = 0.015, and R-square = 0.999. The regression performance indexes of the model revealed that the regression performance of the 1D-CNN model was better, the error between the prediction results and the real section damage angle was approximately 0.9°, and the error was less than 0.5%. The loss curve and prediction results indicate the presence of some overfitting in the model. The results for the regression performance indicators of the MLP model are as follows: RMSE = 1.17, MAPE = 0.014, and R-square = 0.999. The regression performance indexes of the model indicated that the regression performance of the MLP model was excellent, the error between the prediction results and real section damage angle was approximately 1.2°, and the error was less than 0.8%, and its parameter characterization shows that its model prediction performance is not as good as that of the 1D-CNN model.

To clarify whether the model could effectively predict the section damage angle corresponding to the defect echo signal when the guided wave signal at the defect echo was input into the neural network as a feature in the simulation stage, 120 groups of data at the defect echo were extracted from the original simulation data as feature inputs (Figure 9) to the neural network. Following this, the regression effect was compared with that of the entire guided wave signal as a feature input to the neural network.

Table 1 summarizes the final evaluation results for each parameter of the neural network models. In this table, MLP denotes the performance index of the MLP when the full-section guided wave signal is used as the feature input, while 120-MLP represents the performance index of the MLP when the guided wave signal at the defect echo is used as the feature input. Next, 1D-CNN denotes the performance index of the 1D-CNN when the entire guided wave signal is used as the feature input, while 120-1D-CNN denotes the performance index of the 1D-CNN when the guided wave signal at the defect echo is used as the feature input. The numerical results indicate that the prediction performance of the neural networks appears better when the entire guided wave signal is used as the feature input, and the size of the defect is more closely related to the entire guided wave signal. Based on this result, it can be inferred that when only the signal at the echo of the defect is used as the feature input to the network, abundant useful information may be lost, resulting in the inability of the neural network to effectively analyze defects. The 1D-CNN model was more effective in identifying the pipeline damage value from the analog signal.

## 4. Experimental Verification

To verify the effectiveness of the proposed defect quantification method using ultrasonic guided waves in pipes based on neural networks, experimental studies on different crack defect sizes in pipes were conducted. In other words, the neural network method was used to directly predict the sizes of the corresponding pipeline crack defects in the guided wave signal from the entire guided wave signal.

### 4.1. Experimental Design

The instruments and materials used in this study are listed in Table 2 below. Before the experiment, 32 piezoelectric ceramic slices were welded. To ensure that the piezoelectric ceramic slices would remain undamaged under the high welding temperature, the welding temperature was kept below 260 °C during welding. After welding, each piezoelectric ceramic slice was tested using an impedance analyzer to ensure that the frequency interval of the impedance mutation of each piezoelectric ceramic slice remained the same. Generally, after preparing a piezoelectric ceramic slice, treating the pipe to be pasted with this slice is essential. Thus, the cuts at both ends of the pipe were polished using an electric grinder, and the interior and exterior of pipe were cleaned. To place the piezoelectric ceramic slice in close contact with the pipe, it was carefully wiped with a disinfectant alcohol tablet approximately 10 cm from one end of the pipeline to be pasted with the piezoelectric ceramic slice, after which the ceramic piezoelectric slice was pasted evenly and tightly with epoxy resin adhesive. To ensure a clear signal, the excitation and receiving piezoelectric ceramic slices had to be placed at the same axial position as the pipeline and evenly distributed. Sixteen excitation and sixteen receiving piezoelectric ceramic slices were arranged at axial intervals of 5 mm, as depicted in Figure 10. A waveform generator was used to generate a signal modulated by a Hanning window at the excitation end of the pipeline, which was strengthened by a power amplifier and acted on the piezoelectric ceramic slice at the excitation end of the pipeline, to allow the ultrasonic guided wave to traverse all positions along the pipeline. Finally, a time curve of propagation of the ultrasonic guided wave in the pipeline was recorded using an oscilloscope. The actual experiment, instruments, and piping are illustrated in Figure 11.

In the experiment, cutting was carried out first and then measurements were conducted to determine the angle of the section damage. In the absence of damage to the pipeline, the echo signal of the complete pipeline was first obtained, following which the pipeline was cut using an electric friction machine equipped with a special pipeline-cutting blade. After each cut, the crack size was measured and recorded using Vernier calipers (Figure 12). To prevent the cutting-induced temperature rise from affecting guided wave detection, the pipeline was allowed to stand for 20 min after each cutting. After the temperature of the cutting surface was lowered until it was close to the indoor temperature, guided wave detection was performed, and the relevant guided wave data were recorded using an oscilloscope.

To ensure that the excitation wave, defect echo, and the first-end echo could be clearly observed on the oscilloscope, and the guided wave signal could be completely collected, the horizontal scanning time base of the oscilloscope was selected to be 200 μs, and the number of sampling points was 10,000. We selected 31 excitation signals with a central frequency within the range of 50–200 kHz and considered 5 kHz as the optimal step to conduct the frequency sweep operation on the pipeline. During the frequency sweep, we observed that when the central frequency was 80 kHz, the guided wave signal was clear, and no interference was observed from the other modal waveforms, as indicated in Figure 13. Therefore, 80 kHz was selected as the optimal central frequency for the guided wave experiment on this pipe. A total of 21 groups of guided wave data were collected using excitation signals with central frequencies ranging from 70 to 90 kHz, with steps of 1 kHz above and below 80 kHz. To match the numerical simulation, the experiment was terminated when the section damage angle reached 180°. The section damage angles recorded in the experiment are listed in Table 3.

### 4.2. Analysis of Experimental Results

Through the above experimental operation, 10,920 groups of data were collected for neural network training, and 210 groups of guided wave data were collected under optimal frequency conditions. Based on the consistency between the experiment and simulation, the guided wave data at the optimal frequency were analyzed, and the data were divided into training, testing, and verification sets at a ratio of 6:2:2. The MLP and 1D-CNN were used to learn these data, and the regression performance of the two different networks on these data was compared. The final results obtained through network learning and training are as follows.

The training results of the 1D-CNN on the experimental data are shown in Figure 14, and the results of the regression performance indicators of the obtained model are as follows: RMSE = 3.25, MAPE = 0.039, and R-square = 0.993. The regression performance indexes of the model indicate that the regression performance of the 1D-CNN model was not as good as that of the simulated data when the experimental data were used. The error between the predicted results and actual section damage angle was approximately 3.3°, and the error reached 1.8%, which was approximately 1.7 mm when converted into arc length. The results of the regression performance indicators of the MLP are as follows: RMSE = 3.99, MAPE = 0.043, and R-square = 0.990. In general, a neural network can accomplish the quantitative characterization of pipeline damage both from simulation and experimental data, and the regression performance of the 1D-CNN is better than that of the MLP in this task.

Owing to the characteristics of multimodal and frequency dispersions of ultrasonic guided waves, an evident defect echo cannot be observed in the guided wave echo signal under a nonoptimal excitation frequency, as presented in Figure 15. To evaluate whether the neural network could identify ultrasonic guided-wave signal data at other frequencies, 10,920 data groups were input into the network for learning. The corresponding results are presented in Table 4 and Figure 16.

The regression performance indexes of the model indicate that the error between the prediction result of the MLP model and the real section damage angle was approximately 7.13°, and the error reached 3.9%, which was approximately 3.7 mm when converted into arc length. The error between the prediction result of the 1D-CNN model and the real section damage angle was approximately 3.7°, and the error reached 2%, which was approximately 1.9 mm when converted into arc length. These calculation results indicate that a neural network has the ability to identify guided wave echo signal data under nonoptimal excitation frequencies, and it can extract the defect quantity from the data. The 1D-CNN is better in this respect, and its regression performance is basically the same as that of the guided wave signal under the optimal excitation frequency.

Finally, the CNN model was generated, encapsulated, and used to recognize a new dataset formed by integrating and disrupting simulated and experimental data. The recognition results are shown in Figure 17, with performance index scores of RMSE = 12.46, MAPE = 0.241, and R-Square = 0.922. As shown in the figure, the validation data are distributed on both sides of the accurate prediction value. When compared to the validation using only the experimental dataset, the prediction error increases, resulting in an error of 12.46°, up from 7.13°. Additionally, the direction of the arc length also increases by 3. The data have been transformed from the original 2000 groups of validation sets to the current 11,001 groups of validation sets. As a result, the margin of error has increased from 3.7 mm to 6.4 mm. However, this error is still relatively small considering the large amount of accumulated data. These results correspond to an arc length error of 6 mm, further demonstrating the effectiveness of the CNN.

## 5. Conclusions

In this study, the quantitative characterization of pipeline crack defects was realized by combining a 1D-CNN and ultrasonic guided wave technology. Through a numerical simulation and experimental study, a 1D-CNN was used to quantitatively verify the dimension of a pipeline crack defect by constructing an ultrasonic guided wave experimental platform. Ultrasonic guided wave signals with different crack defects in the pipeline were collected and quantitatively analyzed. The primary conclusions of this analysis are as follows:(1)A quantitative analysis of pipeline crack defects can be realized from end to end by combining a 1D-CNN with ultrasonic guided wave technology.(2)Using the entire guided wave signal, including the incident wave, defect echo, and first-end face echo signal, the feature input can improve the accuracy of the 1D-CNN in identifying the size of the pipeline crack defect.(3)The 1D-CNN is more suitable for identification training of the defect size of the pipeline than the MLP.(4)The 1D-CNN can effectively identify the defect size from the echo signal under an excitation signal with different central frequencies, and it can predict the defect size with an error of less than 2%.

In summary, the combination of a 1D-CNN and ultrasonic guided waves can effectively identify pipeline crack damage. Further studies are needed to quantify other types of pipe damage. When the training data samples are insufficient, the CNN tends to produce significant deviations. It is necessary to make breakthroughs in this aspect in the future since the unexplainability of the network makes it impossible to analyze the training data. CNNs can identify various degrees of damage from the response signals of different dispersion states when the data samples are sufficient, which is a unique advantage of CNNs. However, the principle requires in-depth study.

## Figures and Tables

**Figure 1 sensors-24-01204-f001:**
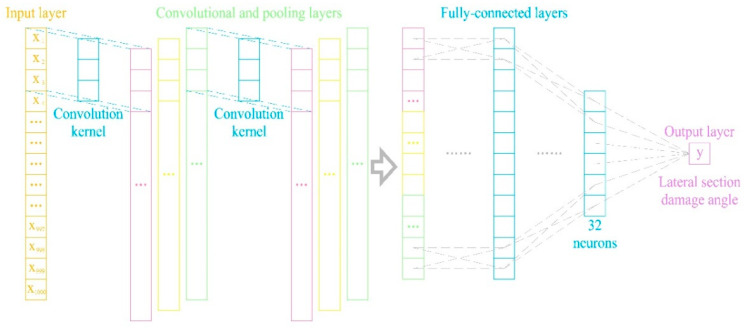
Schematic diagram of a CNN.

**Figure 2 sensors-24-01204-f002:**
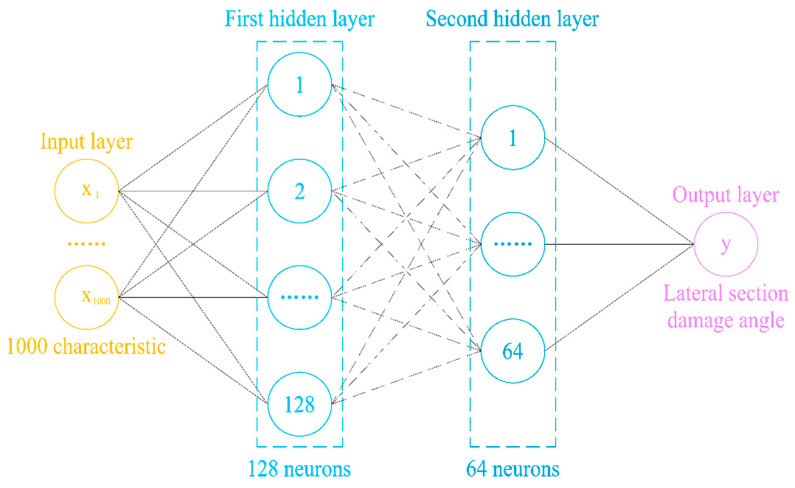
Schematic diagram of an MLP.

**Figure 3 sensors-24-01204-f003:**
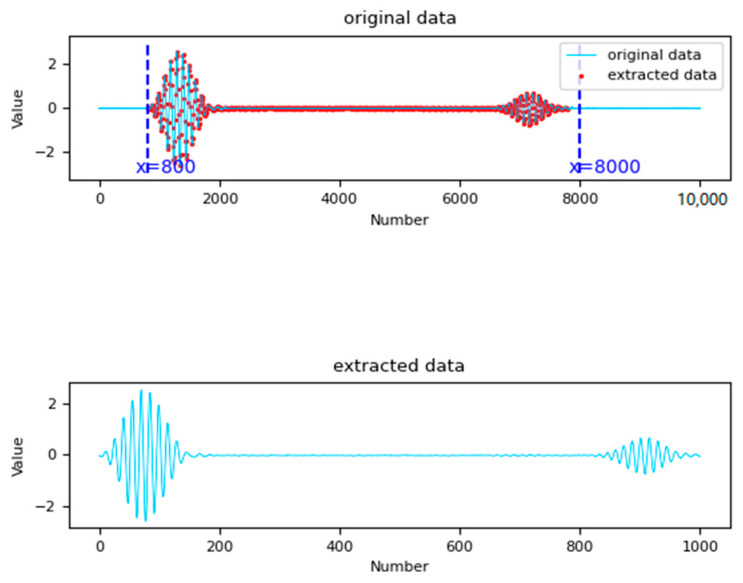
Splitting of experimental data.

**Figure 4 sensors-24-01204-f004:**
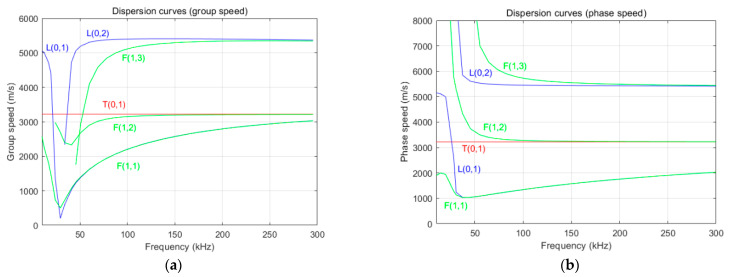
Dispersion curves for (**a**) group speed and (**b**) phase speed.

**Figure 5 sensors-24-01204-f005:**
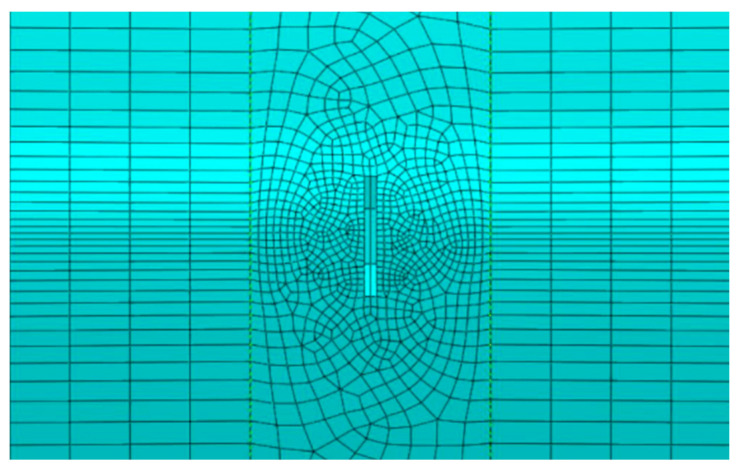
Mesh division with local refinement.

**Figure 6 sensors-24-01204-f006:**
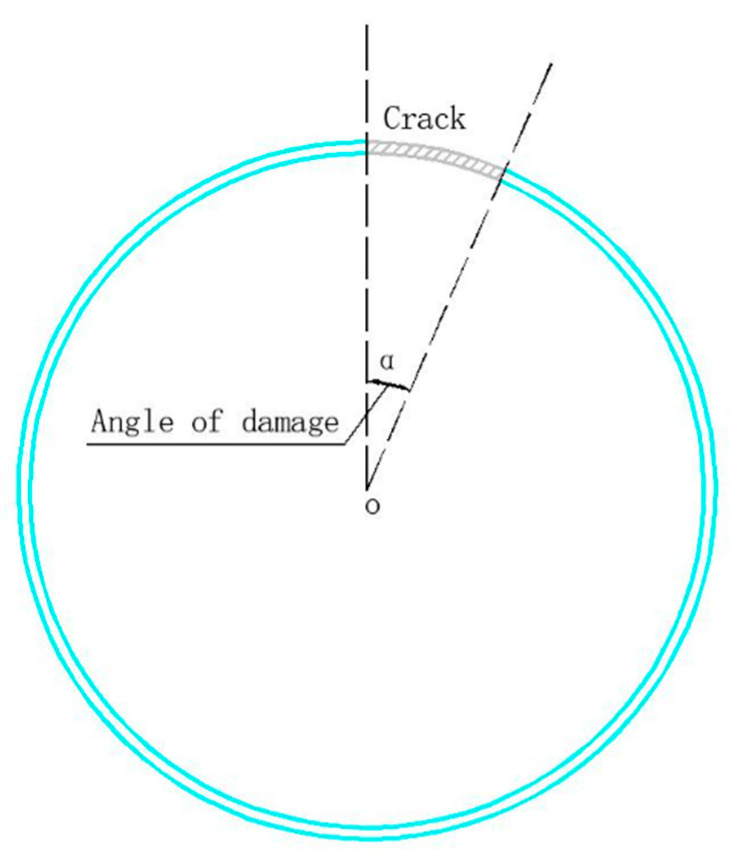
Lateral section damage angle.

**Figure 7 sensors-24-01204-f007:**
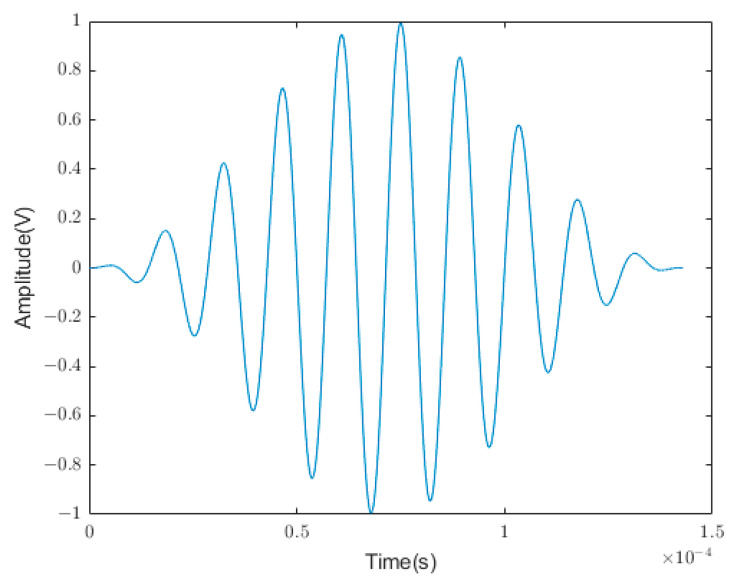
Time-domain signal.

**Figure 8 sensors-24-01204-f008:**
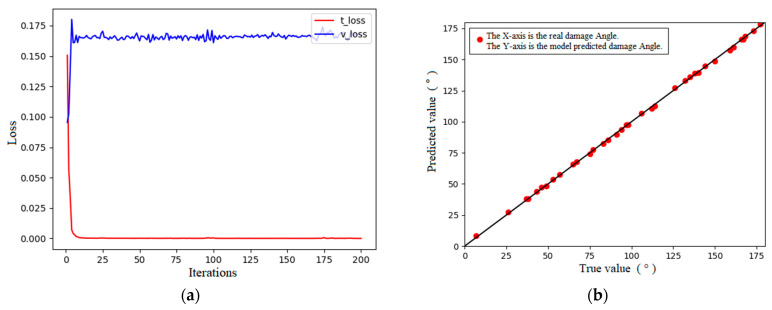
Loss curve and prediction results of the 1D-CNN. (**a**) Loss curves. (**b**) Predicted results.

**Figure 9 sensors-24-01204-f009:**
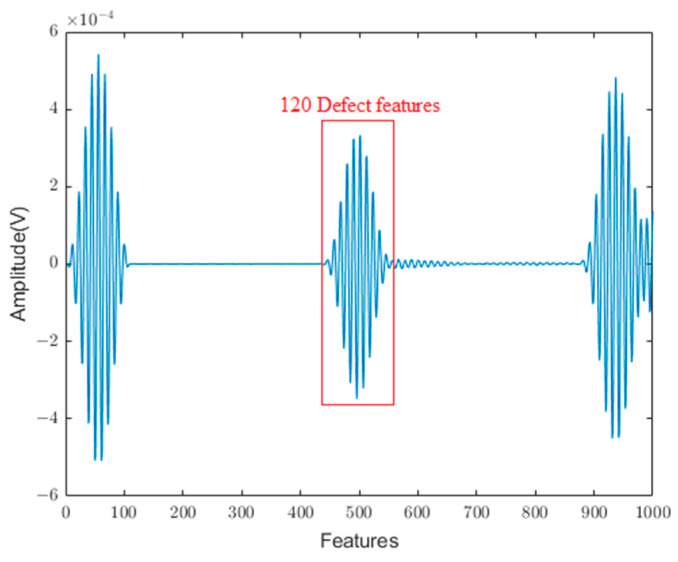
Signal at defect echo.

**Figure 10 sensors-24-01204-f010:**
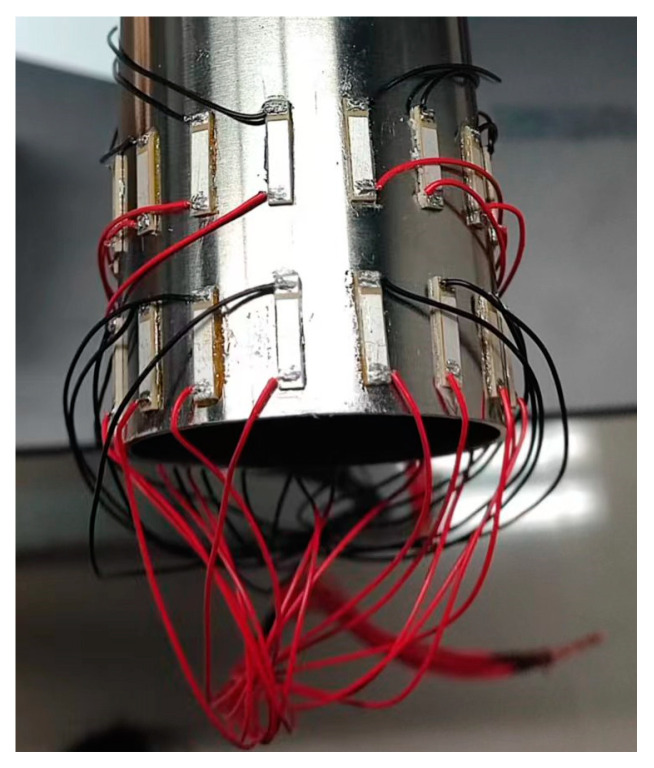
Arrangement of piezoelectric ceramic slice.

**Figure 11 sensors-24-01204-f011:**
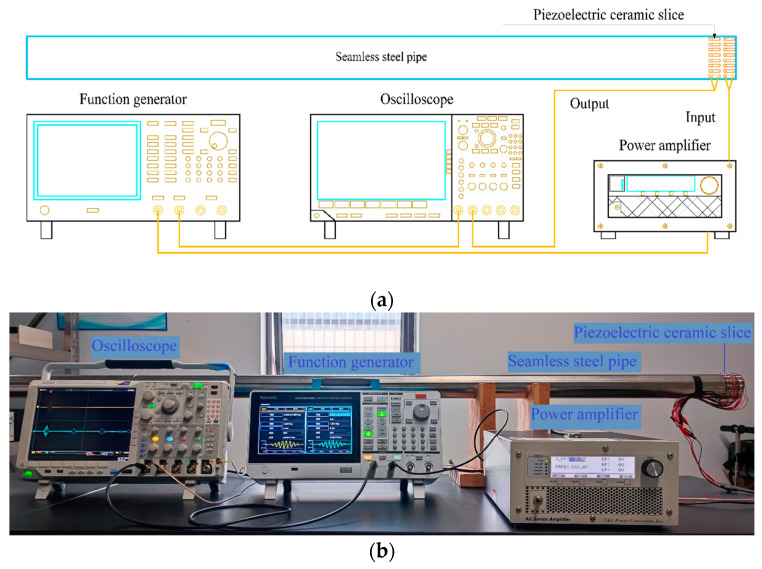
Schematic of the experiment setup. (**a**) Schematic. (**b**) Real object.

**Figure 12 sensors-24-01204-f012:**
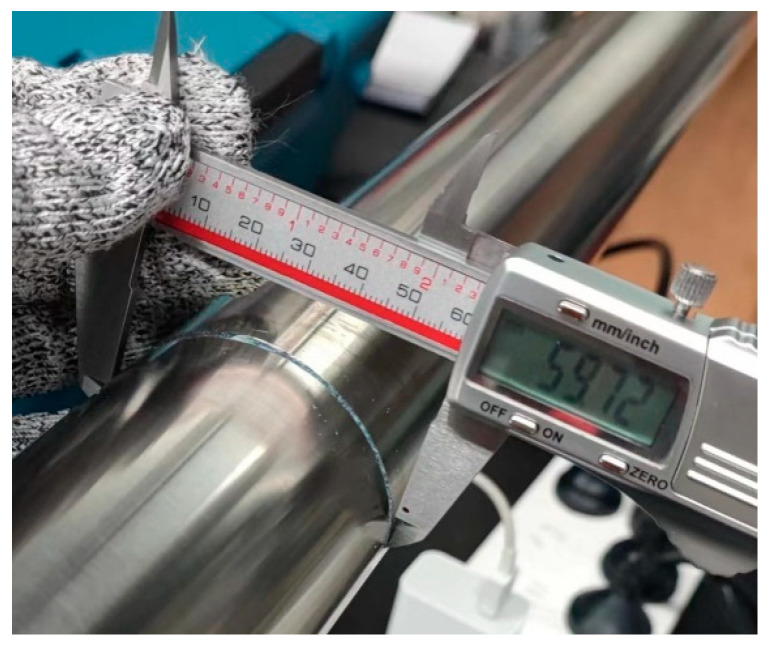
Defect measurement.

**Figure 13 sensors-24-01204-f013:**
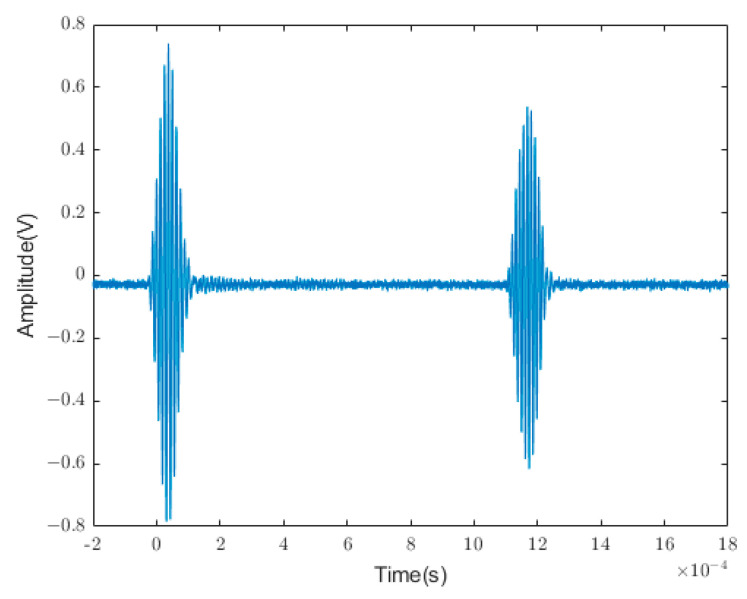
Experimental guided wave data at the optimal central frequency.

**Figure 14 sensors-24-01204-f014:**
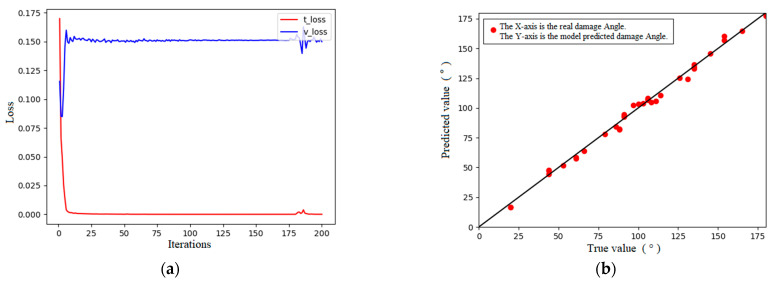
Loss curve and prediction results of the 1D-CNN. (**a**) Loss curves. (**b**) Predicted results.

**Figure 15 sensors-24-01204-f015:**
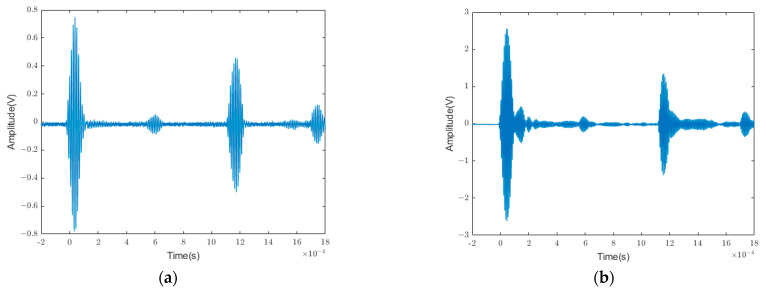
Comparison diagram of ultrasonic guided wave signals. Guided wave echo of an excitation signal with central frequencies of (**a**) 80 kHz and (**b**) 120 kHz.

**Figure 16 sensors-24-01204-f016:**
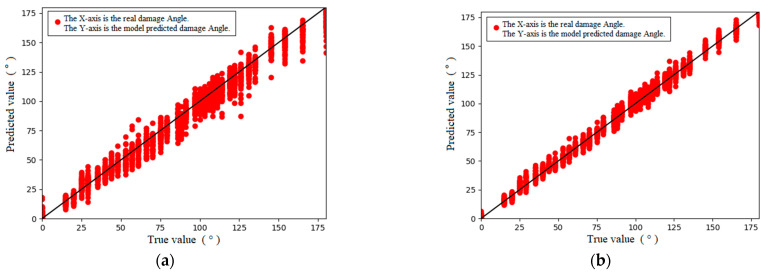
Prediction results. (**a**) MLP; (**b**) 1D-CNN.

**Figure 17 sensors-24-01204-f017:**
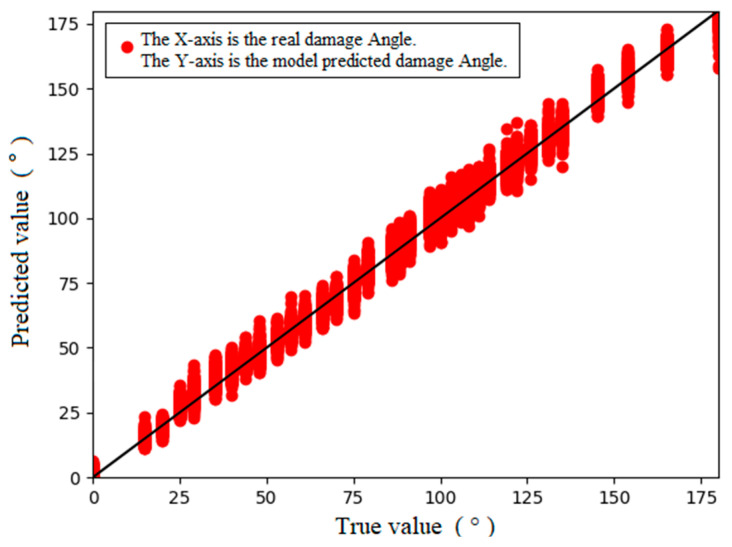
Predicted results for mixed data.

**Table 1 sensors-24-01204-t001:** Performance index evaluation.

Name	RMSE	MAPE	R-Square
MLP	1.17	0.014	0.999
120-MLP	2.45	0.026	0.997
1D-CNN	0.95	0.015	0.999
120-1D-CNN	2.26	0.027	0.998

**Table 2 sensors-24-01204-t002:** Experimental equipment.

Name	Quantity	Type	Notes
Seamless steel pipe	1	Stainless steel	Length of 3 m, outer diameter of 60 mm, wall thickness of 2 mm
Piezoelectric ceramic slice	32	YF3-239-01	Size: 15.5 mm × 3.5 mm × 1 mm
Arbitrary wave function generator	1	AFG31102	Bandwidth: 100 Mhz Sampling rate: 1 GSa/S
Ultrasonic power amplifier	1	AG1020	Frequency: 10 KHz–20 MHz
MDO, mixed domain oscilloscope	1	MDO4054B-3	Bandwidth: 500 Mhz Sampling rate: 2.5 GSa/S
Precision impedance analyzer	1	6632	Error: ±0.08%
Constant temperature welding station	1	SS-257	-
High-precision vernier caliper	1	DYX-DM90150	Precision: 0.01 mm Error: ±0.02 mm
Speed-regulating electric motorcycle	1	DYX-DM7765	Speed: 6000–34,000 RPM/min
Epoxy resin adhesive	Noggin	9911	-

**Table 3 sensors-24-01204-t003:** Experimental data.

Serial Number	The Length of the Damage Tangent (mm)	The Lateral Section Damage Angle (°)	Serial Number	The Length of the Damage Tangent (mm)	The Lateral Section Damage Angle (°)
1	0	0	19	42.62	91
2	8.04	15	20	44.77	97
3	10.31	20	21	45.83	100
4	12.81	25	22	46.82	103
5	14.92	29	23	47.78	106
6	18.08	35	24	48.39	108
7	20.24	40	25	49.25	111
8	22.32	44	26	50.1	114
9	24.1	48	27	51.35	119
10	26.78	53	28	52.21	122
11	28.55	57	29	53.18	126
12	30.2	61	30	54.33	131
13	32.39	66	31	55.09	135
14	34.42	70	32	56.93	145
15	36.42	75	33	58.18	154
16	38.14	79	34	59.23	165
17	40.65	86	35	59.72	180
18	41.53	88			

**Table 4 sensors-24-01204-t004:** Performance index evaluation.

Name	RMSE	MAPE	R-Square
MLP	7.13	0.106	0.974
1D-CNN	3.70	0.071	0.993

## Data Availability

The data generated during this study are currently private due to pending patent applications.

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
