# Peer review of "Quantitative Detection of Pipeline Cracks Based on Ultrasonic Guided Waves and Convolutional Neural Network"

_sensors, 2024, doi:10.3390/s24041204_

Round 1

Reviewer 1 Report (Previous Reviewer 1)

Comments and Suggestions for Authors

In my opinion, the paper has not improved significantly.  At the moment, it is difficult to understand the need for the presence of a modelling part in the paper for the following reasons: 

1. The accuracy of the developed model has not been proven by comparing its results with experimental data. 

2. The impact of using modelling results together with experimental data in the learning set on the accuracy of defect sizing is not considered.

As a result, the paper consists of modelling and experimental parts, which in the current version of the manuscript are weakly related to each other. Developed computer model is not applied to solve the problem of limited data of experimental results, which is one of the factor that limits the application of neural networks in nondestructive testing. Together with the fact that only the crack parameter is considered in the research (other parameters are neglected), limited data could greatly complicate the efficiency of the proposed method in real testing applications.  

Author Response

1. The accuracy of the developed model has not been proven by comparing its results with experimental data. 

Thank you for your question. The paper presents two sets of models: one based on numerical simulation data and the other on experimental data. However, the first model's generalization ability is not perfect, and it cannot accurately identify the experimental dataset or data generated by guided waves with non-optimal excitation frequencies. The first model was abandoned because it could only recognize numerical simulation data. Its role was to provide a preliminary validation of the method's effectiveness during the simulation phase.  However, the second model was more effective as it was able to recognize both numerically simulated and experimental data.

2. The impact of using modelling results together with experimental data in the learning set on the accuracy of defect sizing is not considered.

Thank you for your suggestion. It is true that the paper did not consider the results of the experiments together for comparison. The results of the two experiments have now been compared and added to line 434 of the paper.

the validation data is distributed on both sides of the accurate prediction value. When compared to the validation using only the experimental dataset, the prediction error in-creases, resulting in an error of 12.46°, up from 7.13°. Additionally, the direction of the arc length also increases by 3. The data has been transformed from the original 2000 groups of validation sets to the current 11001 groups of validation sets. As a result, the margin of error has increased from 3.7mm to 6.4mm. However, this error is still relatively small considering the large amount of accumulated data.

 Currently, the model generated by numerical simulation cannot accurately identify experimental data. However, since numerical simulation data is limited, learning with a neural network can lead to overfitting and insufficient generalization ability. Therefore, it is recommended to rely on the model established with experimental data, which accurately identifies the amount of defects. This method enhances the quantitative damage identification of pipelines based on ultrasonic guided waves.

Reviewer 2 Report (New Reviewer)

Comments and Suggestions for Authors

1.      The English of the paper needs revision.

2.      In the introduction, the research gap explanation has not been performed well. What are the advantages of the CNN to the conventional method? Also, there are several papers on the same topic which have been published in this journal. What are the differences in this work?

3.      Gas pipeline damage diagnosis is a supercritical matter. How can the industry trust this method?

4.      You said you used multi-layer in Fig.2. How much is the number of layers? How did you get the optimal number of layers?

5.      How did you validate the Abaqus results?

6.      What is the limitation of the CNN? It should be mentioned in the conclusion.

Comments on the Quality of English Language

1.      The English of the paper needs revision.

Author Response

  1. The English of the paper needs revision.

    Thanks for your suggestions! Some of the English in the paper has been revised!

  2. In the introduction, the research gap explanation has not been performed well. What are the advantages of the CNN to the conventional method? Also, there are several papers on the same topic which have been published in this journal. What are the differences in this work?

    Thanks for your suggestions! The introduction has been revised. See Introduction for details.

    Zhan et al. [7] used a deep learning approach to classify and detect pipe welds in noisy environments. Li et al. [8] used the CNN-LSTM hybrid model to classify pipeline defects. This research has proposed a solution for classifying pipeline defects, but has not thoroughly explored the quantification of pipeline damage. Quantifying pipeline damage often requires a large amount of data to perform regression analysis across the full range of damage, Its impact on simulation and experimental data working conditions diversity is a considerable challenge.

    Compared to the traditional method, CNN can extract the defect loss directly from the response signal of ultrasonic guided waves without the need for complicated pre-processing. This reduces the difficulty of identification technology and prevents the concealment of key information.

  3. Gas pipeline damage diagnosis is a supercritical matter. How can the industry trust this method?

    Thank you for your question. This method can be used as a reference for processing supercritical problems. Simulate guided wave response signals under supercritical environments and train them based on the generated data. If the training results are satisfactory, additional research can be conducted.

  4. You said you used multi-layer in Fig.2. How much is the number of layers? How did you get the optimal number of layers?

    Thank you for your question. Fig. 2 employs two convolutional layers, one pooled layer, and two fully connected layers. The optimal number of layers is determined using the empirical formula, with the optimal convolution kernel size being 3 and the optimal number of convolutional layers being 3. Multiple networks with different architectures are established around this optimal value. Finally, the network architecture with the best training results is selected.

  5. How did you validate the Abaqus results?

    Thank you for your question. To verify the Abaqus results, I first examined whether the model's deformation animation was consistent with the expected deformation pattern. Secondly, I checked whether the guided wave response signal's variation rule under different working conditions was consistent with the defect signal increasing in turn as the damage increases. Finally, when combined with the dispersion curve, the wave velocity is approximately 5 km/s. The pipe length is 3 m, and the point at which the first broken end echo appears is roughly 0.00012 s, which is consistent with the simulation results.

  1. What is the limitation of the CNN? It should be mentioned in the conclusion.

    Thank you for your suggestion. The issue on line 464 has been resolved.

    When the training data samples are insufficient, the CNN network tends to produce significant deviations. It is necessary to make breakthroughs in this aspect in the future since the unexplainability of the network makes it impossible to analyze the training data. CNN can identify various degrees of damage from the response signals of different dispersion states when the data samples are sufficient, which is a unique advantage of CNN. However, the principle requires in-depth study.

Round 2

Reviewer 1 Report (Previous Reviewer 1)

Comments and Suggestions for Authors

In general, all my questions have been addressed. On the one hand, the authors propose the new approach which implies the application of 1D-CNN and the use of time domain signals as input. Experimental verification performed demonstrates the applicability of the developed method. On the other hand, only one parameter of the crack has been considered in the research. This may limit the efficiency of the approach in practice. In addition, the authors did not develop computer models that accurately reproduce the real tests. The inability to increase the training set using computer modelling is another factor that may limit the applicability of the developed approach to practice. 

Reviewer 2 Report (New Reviewer)

Comments and Suggestions for Authors

The paper is acceptable.

This manuscript is a resubmission of an earlier submission. The following is a list of the peer review reports and author responses from that submission.

Round 1

Reviewer 1 Report

Comments and Suggestions for Authors

The article is related to the problem of crack sizing in guided wave ultrasonic testing of pipelines.  In this article, the authors propose to determine the length of cracks using one-dimensional convolutional neural networks. For this purpose, artificial flaws in steel pipelines that replicate a crack with different lengths (expressed in terms of lateral section angle) have been considered. Using computer modeling and real  experiments, a high accuracy of flaw length sizing was reported. Unfortunately, only one parameter of the flaws was considered, although other parameters of the flaws (inclination angle, depth, etc.) also strongly influence the reliability of the pipeline operation.

Authors are invited to respond to the following comments and questions:

1. The accuracy of the results obtained using neural networks is highly dependent on their parameters. However, the details of the implementation of one-dimensional convolutional neural networks and multilayer perceptrons are not discussed in sufficient detail. Therefore, more information should be provided on the implementation of convolutional networks (libraries used, optimizers applied, etc.).

2. One of the limiting factors for the application of deep learning in NDT is the limited amount of inspection data suitable for training artificial neural networks. And one of the solutions considered by different authors is to supplement the training data sets with modeling results. In this paper, the authors perform modeling and real experiments. Did the training data set for defect sizing for signals from the real experiment include the modeling data? If so, how did the modeling results improve the efficiency of the fault sizing solution? If not, why was it not possible to use modeling data together with real results in the training data set? Please consider including this information in your manuscript.

3. There is no data on the depth of the artificial flaws at which they were produced. I suppose this data could also be useful.

Reviewer 2 Report

Comments and Suggestions for Authors

The research investigates the direct utilization of ultrasonic guided wave time-domain signals as input for quantitatively detecting pipeline crack sizes through a one-dimensional convolutional neural network (1D-CNN). The authors conducted numerical simulations and experiments to acquire data, comparing the performance of Multilayer Perceptron (MLP) and 1D-CNN, demonstrating the superior efficacy of the chosen method. While the article exhibits a clear and comprehensive structure with satisfactory experimental results, several issues warrant further elucidation:

1.     The sequence of Figure 2 appearing before Figure 1 in the text should be adjusted.

2.     The description of equal interval sampled signals during simulation lacks precision and detail. Were these 1000 signal sets extracted from different times or from distinct points in the same simulation? What was the interval distance, and what were the respective starting distances? A suggested improvement involves incorporating a schematic diagram for enhanced clarity.

3.     Given the plethora of data-driven classification methods available, what motivated the selection of Multilayer Perceptron (MLP) for comparison with CNN?

4.     On line 308, it is recommended to replace "AB glue" with "epoxy resin."

5.     How were the training epochs for the neural network determined?

6.     After establishing hyperparameters through formulae, were there subsequent experiments to verify their optimality?

7.     What are the comparative results when employing traditional methods (e.g., feature extraction for training) with the measured data? Specifically, in what aspects does the selected method outperform traditional approaches?

Comments on the Quality of English Language

The manuscript can be better if the authors can pay attention to the correctness of proper nouns

Reviewer 3 Report

Comments and Suggestions for Authors

In the paper, the author introduces the research work on quantitative detection of pipeline cracks using convolutional neural networks with ultrasonic guided waves. Here are the comments for the authors:

1. The author attempts to use convolutional neural networks as the detection model and ultrasonic guided wave response as the dataset to quantitatively detect pipeline cracks and evaluate status information. This approach will face many problems in practical applications, such as the different dispersion of different structures, resulting in different ultrasonic guided wave responses. Secondly, the occurrence of crack defects usually is random, and their orientation also varies, resulting in complex changes in the responses of ultrasonic guided waves. Therefore, the method proposed by the authors may encounter many problems in practical application and its feasibility is not satisfactory;

2. Here are some specific review comments:

Why directly use ultrasonic guided wave response signals instead of characteristic parameters? The latter has a stronger correlation with defects.

What is the reason and basis for the arrangement of the piezoelectric array as shown in Figure 9?

How can this method be applied to irregular structures? This type structure is widely used, and its relatively complex structural forms can lead to the complexity of ultrasonic guided wave response.